# Evaluation of Cisplatin-Induced Pathology in the Larval Zebrafish Lateral Line

**DOI:** 10.3390/ijms232214302

**Published:** 2022-11-18

**Authors:** David S. Lee, Angela Schrader, Emily Bell, Mark E. Warchol, Lavinia Sheets

**Affiliations:** 1Department of Otolaryngology—Head and Neck Surgery, Washington University School of Medicine, St. Louis, MO 63110, USA; 2Department of Neuroscience, Washington University School of Medicine, St. Louis, MO 63110, USA; 3Department of Developmental Biology, Washington University School of Medicine, St. Louis, MO 63110, USA

**Keywords:** cisplatin, ototoxicity, zebrafish, lateral line organ, macrophage, rheotaxis

## Abstract

Cisplatin is an effective anticancer agent, but also causes permanent hearing loss by damaging hair cells—the sensory receptors essential for hearing. There is an urgent clinical need to protect cochlear hair cells in patients undergoing cisplatin chemotherapy. The zebrafish lateral line organ contains hair cells and has been frequently used in studies to screen for otoprotective compounds. However, these studies have employed a wide range of cisplatin dosages and exposure times. We therefore performed a comprehensive evaluation of cisplatin ototoxicity in the zebrafish lateral line with the goal of producing a standardized, clinically relevant protocol for future studies. To define the dose- and time-response patterns of cisplatin-induced hair-cell death, we treated 6-day-old larvae for 2 h in 50 µM–1 mM cisplatin and allowed them to recover. We observed delayed hair cell death, which peaked at 4–8 h post-exposure. Cisplatin also activated a robust inflammatory response, as determined by macrophage recruitment and phagocytosis of hair cells. However, selective depletion of macrophages did not affect hair cell loss. We also examined the effect of cisplatin treatment on fish behavior and found that cisplatin-induced lateral line injury measurably impaired rheotaxis. Finally, we examined the function of remaining hair cells that appeared resistant to cisplatin treatment. We observed significantly reduced uptake of the cationic dye FM1-43 in these cells relative to untreated controls, indicating that surviving hair cells may be functionally impaired. Cumulatively, these results indicate that relatively brief exposures to cisplatin can produce hair cell damage and delayed hair cell death. Our observations provide guidance on standardizing methods for the use of the zebrafish model in studies of cisplatin ototoxicity.

## 1. Introduction

Cisplatin is an antineoplastic drug that has been a mainstay chemotherapy for a wide range of solid malignancies since the 1970s [1,2]. While effective as a therapy, cisplatin also causes permanent hearing loss in a large percentage of patients [3,4]. Cisplatin-induced hearing loss is progressive, bilateral (i.e., affects both ears), and permanent. This hearing loss impacts patients’ quality of life and is particularly detrimental to pediatric patients, who are more susceptible to cisplatin-induced hearing loss and experience negative impacts on language development and social skills that have long-term repercussions [5]. As it is unlikely that cisplatin will be completely replaced by other chemotherapeutic agents in the foreseeable future, identifying ways to mitigate cisplatin ototoxicity is of high clinical importance.

Cisplatin irreversibly damages cochlear hair cells [6]. Hair cells are the sensory receptors of the auditory and vestibular systems of all vertebrates. While the overall cellular mechanisms for cisplatin-induced hearing loss are not entirely understood, it is known that cisplatin damages cochlear tissue, activates an inflammatory response, and contributes to apoptotic death of sensory hair cells [7]. Preventing hair cell death from cisplatin to preserve hearing is particularly important, as hair cells in the mammalian inner ear do not spontaneously regenerate [8].

Zebrafish have been used in many studies as an in vivo model to evaluate the cellular mechanisms of cisplatin-induced hair cell death and to identify potential protective compounds that mitigate cisplatin ototoxicity [9,10,11,12,13,14,15]. They are attractive models for studying cisplatin injury because zebrafish possess a network of sensory organs on the surface of their bodies called the lateral line. The lateral line is used to detect water flow and is comprised of structures called neuromasts, which contain clusters of innervated hair cells and associated supporting cells. Neuromast hair cells share a high degree of conserved physiology with mammalian cochlear hair cells and are functionally mature at 5 days post fertilization (dpf) [16]. Use of zebrafish larvae in ototoxicity research has many advantages, including rapid development, superficial location of hair cells, and larval transparency, which provides the ability to monitor dynamic cellular changes in vivo. Furthermore, they may be spawned at scale to support high-throughput screening for targeted drug discovery [17].

While zebrafish larvae have been used in numerous studies to examine the mechanisms of cisplatin ototoxicity and evaluate strategies to protect hair cells from cisplatin, it has been challenging to integrate the results of these studies into a comprehensive model of cisplatin-induced hair cell damage. This challenge is in large part due to a lack of a consensus protocol for cisplatin exposure and of standardized methods of outcome measurements [18]. We therefore performed an evaluation of cisplatin ototoxicity in the zebrafish lateral line with the goal of producing a standardized protocol for future studies. Our observations revealed that relatively brief exposures to cisplatin initiated delayed hair cell death and activated an inflammatory response. Lateral-line mediated behavior was also impaired following cisplatin exposure, and hair cells that survived cisplatin exposure showed significantly decreased FM1-43 uptake, indicating reduced hair cell mechanotransduction. Cumulatively, these results suggest a standardized zebrafish protocol for cisplatin toxicity that involves shorter exposure times, evaluating hair cell numbers following an 8–24 h recovery time, and monitoring function in surviving hair cells when evaluating the effectiveness of otoprotective drugs.

## 2. Results

### 2.1. Cisplatin Exposure Initiates Progressive Lateral Line Hair Cell Loss

A previous study that established zebrafish as a model for cisplatin ototoxicity reported both dose and time of exposure independently influence cisplatin-induced hair cell loss [11]. While it is known that exposure to lower doses of cisplatin for longer amounts of time are as ototoxic as high doses for shorter amounts of time, the precise dose-time relationship for hair cell damage is not known. To better define the dose–response pattern of cisplatin-induced hair-cell death in zebrafish lateral line organs, we exposed 6-day-old zebrafish to increasing concentrations of cisplatin for 2 h (Figure 1A). We then labeled hair cell nuclei with DAPI after either 2- or 24 h recovery in embryo medium (EM) to examine hair-cell survival (Figure 1B–E). Following 2 h of recovery, we observed loss of neuromast hair cells in larvae exposed to high concentrations of cisplatin (500 µM and 1 mM), but no observable hair cell loss in larvae exposed to lower concentrations (50–250 µM; Figure 1D). However, when we examined larvae at 24 h after cisplatin exposure, we observed dose-dependent loss of hair cells in neuromasts exposed to all concentrations of cisplatin (Figure 1C–E). We subsequently evaluated the time course of hair cell loss in fish exposed to a moderate dose of cisplatin (250 µM). We chose this concentration because it was the highest dose of cisplatin without an appreciable lesion after 2 h of recovery, enabling the greatest dynamic range over this time period. Maximal hair cell loss appeared to occur between 4–8 h post cisplatin exposure (Figure 1F). These observations indicate that relatively short exposures to cisplatin initiates progressive hair cell loss in zebrafish lateral line organs, but that hair cell death can require several hours to manifest.

### 2.2. Cisplatin Treatment Induces an Inflammatory Response That Corresponds with the Timing of Neuromast Hair Cell Death 

Injury to the cochlea by noise or ototoxic drugs induces an inflammatory response, including the recruitment of macrophages to injured cochlear tissue [19]. In previous studies, we have also observed rapid recruitment of macrophages to lateral line neuromasts following injury caused by excess mechanical stimulation or neomycin exposure [20,21,22]. We therefore evaluated the time course of macrophage recruitment to neuromasts following cisplatin injury. These experiments utilized transgenic larvae that stably express YFP in their macrophages (w200Tg; *Tg(mpeg1:YFP)*). Six-day-old transgenic larvae were treated with 250 µM cisplatin for 2 h, then allowed to recover for 2, 4, 8 or 24 h (Figure 2A–D). We observed no increase in the number of macrophages within 25 µm of injured neuromasts (Figure 2E), which is comparable to what we previously observed in in zebrafish exposed to neomycin [21]. We also saw no significant increase in microphage contacts with injured neuromasts at fixed time points following cisplatin exposure (Figure 2F), which contrasts with the significant increase in macrophage contacts observed in injured neuromasts following neomycin exposure. Nevertheless, we observed activation and phagocytic activity of macrophages in response to cisplatin-induced hair cell death. Phagocytosis of hair cell debris increased and peaked 4 h following exposure (Figure 2G), which corresponds to the time course of hair cell loss occurring 4–8 h following 250 µM cisplatin (Figure 1F). It is worth noting that hair cell death following neomycin exposure is rapid, with hair cell death and subsequent macrophage response occurring in <30 min [21]. These data may reflect a limitation of capturing macrophage dynamics over the more delayed time course of hair cell death following cisplatin.

### 2.3. Inflammation Does Not Enhance Cisplatin Ototoxicity

In addition to clearing cellular debris from damaged tissues, macrophages may influence the pathogenesis of sensory cells by secreting proinflammatory cytokines that lead to the production to reactive oxygen species and apoptosis of damaged cells [23]. To evaluate the role of macrophages in cisplatin-induced hair cell death, we quantified neuromast hair cell loss in fish where macrophages were selectively ablated. Experiments used double transgenic fish (gl25Tg, *Tg(mpeg1.1:GAL4FF*); c264Tg, *Tg(UAS-E1B:NTR-mCherry*) that have macrophage-specific expression of the bacterial enzyme nitroreductase (NTR). When exposed to the nontoxic prodrug metronidazole (MTZ), NTR-expressing cells convert MTZ to a toxic form that induces rapid destruction of NTR-expressing cells [24]. To verify macrophage ablation following MTZ treatment, we quantified the number of macrophages in the posterior-most 500 µm of the tail and observed significant depletion of macrophages in MTZ-treated larvae, relative to siblings treated with drug-carrier alone (Figure 3A–C). We then exposed both untreated and MTZ-treated larvae to 250 µM cisplatin for 2 h followed by 24 h recovery (Figure 3D,E). We observed similar hair cell loss in both macrophage-depleted and control larvae (Figure 3F), suggesting that macrophages do not contribute to cisplatin ototoxicity in zebrafish lateral line organs. 

### 2.4. Cisplatin Treatment Impacts Rheotaxis Behavior and Residual Hair Cell Function

Research using zebrafish to evaluate cisplatin ototoxicity usually assesses neuromast damage by quantifying the number of hair cells remaining at short time intervals after cisplatin exposure [18]. While behavioral assays have not been extensively used to determine damage to lateral line organs, a previous study indicated that cisplatin-induced neuromast damage was correlated with reduced positive rheotaxis—a swimming behavior in which fish orient and swim against water flow [25]. We evaluated positive rheotaxis behavior in fish that were exposed to 250 µM and 1 mM cisplatin, in order to compare the impact of partial (~60% hair cell loss, Figure 1E) and nearly complete (~80% hair cell loss, Figure 1E) lesions on this behavior. Our studies used a custom-built apparatus that presents laminar flow of embryo media at a rate of 5 mm/s and records larval swimming behavior in the absence of visual cues (Figure 4A; [26]). Fish were tested at 24 h after a 2 h exposure to either 250 µM or 1 mM cisplatin. We have previously reported that zebrafish with intact lateral line function are able to ‘station hold’ and maintain their position at the source of the water flow stimulus (where the intensity of flow is the strongest), while fish lacking lateral line input following treatment with neomycin or CuSO_4_ can orient against water flow but cannot maintain position near the flow source [26]. Fish treated with cisplatin displayed similar deficits in rheotaxis behavior. Control larval zebrafish were generally found near the source of the water flow during the 10 s stimulus (Figure 4B), while fish treated with either moderate or high dose cisplatin were generally found swimming toward the back of the arena where the water flow was the weakest (Figure 4C,D). These data qualitatively corresponded to recordings of fish behavior; fish with intact lateral line organs performed characteristic burst-and-glide activity to station hold at the front of the microflume, while lesioned fish struggled to execute counterflow movements (Appendix A). Cumulatively, these observations indicate that moderate and high dose cisplatin impede zebrafish ability to station hold and orient against water flow, possibly due to functional impairment of the lateral line organ.

The observation that rheotaxis behavior was impaired following treatment with 250 µM cisplatin was notable, as we typically find that ~30% of neuromast hair cells survive after cisplatin treatment at this dose and recovery time (Figure 1E,F). Given that rheotaxis behavior was similarly impaired following exposure to either moderate or high dose cisplatin, we then wanted to determine whether the surviving hair cells following more moderate cisplatin treatment were functional. Previous studies have established that hair cell uptake of the cationic dye FM1-43 can be used to quantify hair cell mechanotransduction [27,28]. Fish that were exposed to cisplatin and allowed to recover for 24 h were briefly co-incubated in DAPI and FM1-43 to evaluate hair cell viability and mechanotransduction, respectively (Figure 5A–C). While we observed significantly more hair cell loss per neuromast with high dose (1 mM) relative to moderate (250 µM) cisplatin (Figure 5E), we saw similar reductions (~40%) of FM1-43 uptake in surviving hair cells with both cisplatin doses, relative to untreated controls (Figure 5D; stats here). Cumulatively, these results suggest that relatively brief exposures to cisplatin produce both delayed hair cell death and damage to remaining hair cells that impact lateral line organ function.

## 3. Discussion

Zebrafish lateral line organs are a well-established in vivo model for ototoxicity research and provide many experimental advantages for evaluating the effects of drugs on sensory hair cells [29]. A recent systematic review analyzed 26 studies performed between 2009–2020 that used the zebrafish lateral line to evaluate cisplatin-induced ototoxicity and otoprotective compounds [18]. While this report confirmed the usefulness of the zebrafish model for understanding the cellular mechanisms of cisplatin-induced hair cell damage, it concluded that this scientific field should prioritize the development of a consensus protocol incorporating standardized ranges of cisplatin concentrations and durations of exposure. In this study, we evaluated cisplatin-induced lateral line damage and hair cell loss in 6–7-day-old larval zebrafish with the goal of producing such a standardized protocol. Our data revealed that relatively brief cisplatin exposures induced delayed hair cell death that peaked 4–8 h following treatment. Further, we examined the effect of cisplatin exposure on lateral-line mediated behavior and hair cell function 24 h following moderate and high dose cisplatin exposure. We observed impaired rheotaxis behavior and significantly reduced hair cell FM1-43 uptake in both treatment conditions, indicating that cisplatin can cause dysfunction in hair cells that have survived for 16–20 h after the period of maximal hair cell loss.

Previous cisplatin ototoxicity studies in zebrafish have generally examined hair cell survival immediately following prolonged (6–24 h) cisplatin exposure [18]. An initial study exposed larval zebrafish for 2–12 h to cisplatin (at concentrations of 50 µM–1 mM) and then quantified hair cell survival immediately after exposure [11]. Continuous treatment for 8–12 h with high doses (500 µM–1 mM) of cisplatin led to nearly complete neuromast hair cell loss. Most subsequent zebrafish studies of cisplatin ototoxicity have employed similar high dose/long exposure protocols. Notably, Ou et al. also assessed the progression of hair cell loss; following 2 h exposure to 1 mM cisplatin, fish that underwent a 4 h recovery period experienced greater hair cell loss than those evaluated immediately after cisplatin exposure [11]. Our present results expand on these early observations and demonstrate progressive hair cell loss following 2 h cisplatin exposure at all doses examined. The time course of hair cell death also corresponded with macrophage clearance of hair cell debris that peaked 4 h following cisplatin exposure. Cumulatively, these observations support that hair cell death, even following high doses of cisplatin, is progressive. Moreover, shorter time exposure to cisplatin will further benefit studies by limiting off target effects, as extended exposure of whole animals to cisplatin may cause off-target toxic effects that could complicate mechanistic studies. Therefore, we propose that future investigations of cisplatin ototoxicity in zebrafish larvae use a standardized 2 h exposure protocol laid out in Figure 1A to minimize cisplatin’s non-specific toxicities while still capturing maximal hair cell loss. This protocol uses a broad range of cisplatin concentrations (50 µM to 1 mM) and recovery periods (2 to 24 h) to accommodate different research questions, with corresponding dose- and time-response curves in Figure 1D–F as guides. Such a standard exposure protocol may facilitate comparisons across different studies of cisplatin ototoxicity and otoprotective compounds.

Evaluating cisplatin ototoxicity in the lateral line of larval zebrafish and testing the efficacy of protective compounds typically relies on morphometric analysis of neuromasts immediately following cisplatin exposure, via either fluorescent vital dyes or immunohistochemical labeling of hair cells [18]. While scoring morphological integrity or quantifying the number of intact hair cells shows the degree of anatomical damage to neuromasts, it does not provide information on whether hair cells that survive cisplatin treatment retain normal function. Our results indicate that both lateral line-mediated behavior and hair cell mechanotransduction are impaired at 24 h after cisplatin exposure; a time point when maximum hair cell death has likely occurred but also insufficient time for any significant hair cell recovery via regeneration (Figure 1F). We observed impaired rheotaxis behavior in larvae that were exposed to moderate and high doses of cisplatin, suggesting that lateral-line function in these fish was compromised. The underlying reasons for this behavioral deficit are not clear. While it is possible that the numbers of hair cells per neuromast that survived cisplatin exposures were not sufficient to provide accurate data on water flow to the regions of the brain that mediate rheotaxis behavior, previous studies suggest that only a subset of neuromast hair cells are needed for proper lateral line function [25,30]. Alternatively, our observation that surviving hair cells show reduced uptake of FM1-43 relative to untreated control suggests that cisplatin may inflict damage to surviving sensory cells that impacts their ability to detect water flow. Non-lethal changes in hair cells caused by exposure to cisplatin have not been extensively studied, but our results suggest that they occur. Still undefined are what aspects of hair cell physiology are targeted following cisplatin exposure. We have previously reported that cisplatin disrupts mitochondrial bioenergetics in zebrafish lateral line hair cells [31] and it is possible that such metabolic stress might have downstream consequences on sensory function. For example, reduction of ATP-mediated ion transport may lead to a reduction (i.e., depolarization) of a hair cell’s resting potential, which would then reduce the driving force operating on external cations, leading to a reduction in transduction current. In summary, these observations suggesting neuromast dysfunction following cisplatin treatment and recovery appear relevant to the design of studies that use zebrafish larvae to assess whether compounds can provide protection from cisplatin ototoxicity. Consequently, we encourage evaluation of residual hair cell function following cisplatin exposure in zebrafish to complement morphometric analysis in a similar way that distortion product otoacoustic emissions and auditory brain response experiments support morphometric analysis in the murine model. In the context of novel drug discovery, we speculate that performing morphometric hair cell analysis as a screening readout for otoprotective compounds and then corroborating positive results with a functional assay may increase successful translation to other preclinical models.

One further unresolved issue is whether cisplatin-induced hair cell damage is permanent or reversible. Recent studies support that mitochondrial dysfunction precedes hair cell apoptosis following cisplatin exposure and modulating mitochondrial dynamics may provide some protection against cisplatin induced hair cell death [31,32]. Surviving hair cells may have mitochondrial damage that impairs hair cell function but does not progress to initiation of apoptosis; reduced FM1-43 uptake in neuromast hair cells has been observed in fish with a mutation affecting mitochondrial homeostasis [33], and those data are comparable to what we measured following cisplatin exposure and recovery (Figure 5A–D). It is not known whether cisplatin-induced mitochondrial damage may be repaired over time or eliminated by mitochondrial replication. In addition, it is possible that mitochondrial DNA may be directly damaged by cisplatin, leading to both acute and permanent deficits in mitochondrial function. Future studies should explore whether surviving hair cell fully recover function and, if so, whether potential therapies can be targeted to promote hair cell repair.

## 4. Materials and Methods

### 4.1. Ethics Statement

This study was performed with the approval of the Institutional Animal Care and Use Committee of Washington University School of Medicine in St. Louis (Protocol Number: 20-0158) and in accordance with NIH guidelines for use of zebrafish.

### 4.2. Zebrafish Husbandry and Lines

Adult zebrafish were maintained in group housing and standard conditions at the Washington University Zebrafish Facility. Embryos were maintained in embryo media (EM: 15 mM NaCl, 0.5 mM KCl, 1 mM CaCl_2_, 1 mM MgSO_4_, 0.15 mM KH_2_PO_4_, 0.042 mM Na_2_HPO_4_, 0.714 mM NaHCO_3_) at 28 °C with a 14 h light and 10 h dark cycle. Beginning at 4 days post-fertilization (dpf), larvae were raised in 100–200 mL EM in 250-mL plastic beakers and fed rotifers daily. Experiments were started in the mid-morning and completed by the late afternoon. The sex of the animal was not considered in our experiments because sex cannot be predicted or determined in zebrafish larvae.

The following fish lines were used in this study: AB, *Tg(mpeg1:YFP)w200* (w200Tg; [34]), *Tg(mpeg1:Gal4FF)gl25* (gl25Tg; [35]), and *Tg(UAS-E1B:NTR-mCherry)* (c264Tg). Transgenic larval zebrafish were screened for transgenic fluorophores at 3–5 dpf under sedation with 0.01% tricaine in EM using a Leica MZ10 F stereomicroscope with fluorescence (Leica Microsystems, Wetzlar, Germany) equipped with a GFP and a DSR filter set (Chroma Technology, Bellows Falls, VT, USA).

### 4.3. Larval Zebrafish Exposure to Cisplatin

Free-swimming larvae were treated with cisplatin at 6 dpf. Groups of ~20–30 larvae were placed in 25 mm cell strainers (Corning Cell Strainer, Corning, NY, USA) and incubated for 2 h in 30 mL of EM containing 0.1% dimethyl sulfoxide (DMSO) with 50 µM–1 mM cisplatin (Abcam, Cambridge, UK) at 28 °C. While DMSO increases cisplatin potency and is not a required carrier for intracellular uptake [36], zebrafish are commonly used to screen for otoprotective drugs, so experiments were designed to be generalizable to otoprotective drug studies. The exposure time for cisplatin of 2 h was evaluated and described in a previous study from our group [31]. Zebrafish were then rinsed and allowed to recover in 30 mL of EM for 2–24 h. Assays were not performed beyond 24 h of recovery to capture phenotypic changes prior to spontaneous zebrafish hair cell regeneration.

### 4.4. Selective Depletion of Macrophages

The possible influence of macrophages on cisplatin ototoxicity was examined using double transgenic fish (gl25Tg, *Tg(mpeg1.1:GAL4FF)*; c264Tg, *Tg(UAS-E1B:NTR-mCherry*). To eliminate NTR-expressing macrophages, 6 dpf double transgenic larvae were incubated for 24 h in either 10 mM MTZ (Sigma Aldrich, Darmstadt, Germany; with 0.1% DMSO) or 0.1% DMSO alone. Larvae were then rinsed in EM and subsequently treated with 250 µM cisplatin or 0.1% DMSO for 2 h as described above. Larvae were then allowed to recover in EM for 24 h prior to evaluating cisplatin ototoxicity and macrophage depletion.

### 4.5. Immunohistochemical Labeling

Prior to fixation, free swimming larvae (6–7 dpf) were pretreated with DAPI diluted in EM for 4 min to label neuromast hair cell nuclei (1:2000; Sigma Aldrich, Darmstadt, Germany) as previously described [37]. DAPI selectively entered hair cells through their mechanoelectrical transduction (MET) channels, which are large mechanically gated ion channels unique to hair cell that open upon deflection. Larvae were then rinsed in fresh EM, briefly sedated on ice, transferred to cold fixative (4% paraformaldehyde in 0.1 M phosphate buffer (PBS)) in 1.5 mL Eppendorf tubes, and fixed overnight at 4–8 °C. Following fixation, fish were thoroughly rinsed in PBS. Nonspecific antibody binding was blocked by treatment for 2 h in 5% normal horse serum (NHS; Sigma Aldrich, Darmstadt, Germany) in phosphate-buffered saline (PBS: 8 g NaCl, 0.2 g KCl, 0.24 g KH_2_PO_4_, 14.4 g Na_2_HPO_4_ per 1 L distilled H_2_O) with 1% Triton X-100. This was followed by incubation in primary antibodies, which were diluted in PBS with 2% NHS and 1% Triton X-100. All specimens were treated in antibody solutions overnight, at room temperature and with mild agitation The next day, specimens were rinsed 3 times in PBS and incubated for 2 h in secondary antibodies (anti-rabbit IgG and anti-mouse IgG) were conjugated to Alexa-488 and Alexa-555, respectively (1:500, Invitrogen, Waltham, MA, USA). Following thorough rinsing in PBS, fish were mounted in glycerol:PBS (9:1) on microscope slides. Specimens were then coverslipped and sealed with clear nail polish. 

### 4.6. Primary Antibodies

Hair cells were labeled with HCS-1, which is specific for otoferlin. HCS-1 is an IgG2a mouse monoclonal and was obtained from the Developmental Studies Hybridoma Bank (University of Iowa) as a purified concentrate and used at 1:500 dilution. The YFP or mCherry signals in macrophages were amplified by labeling with either anti-GFP or anti-mCherry (1:500, ThermoFisher, Waltham, MA, USA).

### 4.7. Confocal Imaging of Fixed Samples

Images of fixed samples were acquired using an LSM 700 laser scanning confocal microscope with a 63×/1.4 N.A. Plan-Apochromat oil-immersion objective (Carl Zeiss, Jena, Germany). Confocal stacks of 15 µm depth were collected with a z-step of 1 µm. This study focused on posterior lateral line neuromasts L3, L4, and L5, as defined in [38]. Regardless of the fluorophore expressed (either YFP or mCherry), macrophages in all figure images were pseudo-colored cyan.

### 4.8. Confocal Image Processing and Analysis

Confocal image stacks were reconstructed and analyzed using Volocity software (Quorum Technologies, Lewes, UK). For hair cell counts, an intact hair cell was defined as a normal-looking DAPI-stained nucleus, surrounded by uninterrupted otoferlin immunoreactivity. Pyknotic nuclei were identified as condensed, intensely bright DAPI labeled puncta. Macrophage activity was quantified by scrolling through image stacks (in the z-dimension) and counting the number of macrophages within 25 µm radius of a neuromast (using a circle inscribed on the confocal image of a particular neuromast). The number of macrophages contacting a neuromast was determined by scrolling through the x-y planes of each image stack (1 µm interval between x-y planes, 15 µm total depth) and the counting macrophages that were in direct contact with intact hair cells. Finally, the number of macrophages that had internalized Otoferlin-labeled material (hair cell debris) were counted and were assumed to reflect the number of phagocytic events. For each metric, the recorded number reflected the activity of a single macrophage, i.e., a macrophage that made contacts with multiple hair cells and/or had internalized debris from several hair cells was still classified as a single ‘event.’ Subsequent image processing for display within figures was performed using Photoshop and Illustrator software (Adobe, Mountain View, CA, USA).

### 4.9. Rheotaxis Behavioral Assay and Analysis

Rheotaxis is a swimming behavior characterized by orienting against the direction of water flow and station holding through counterflow burst-and-glide movements. It incorporates sensory input from the lateral line, vestibular, visual, and mechanotactile systems. To investigate the effect of cisplatin injury to the lateral line organ on rheotaxis performance, a rheotaxis behavioral assay was performed as previously described [26]. Briefly, groups of ~20–30 wildtype larvae (6 dpf) were placed in 25 mm cell strainers (Corning Cell Strainer, Corning, NY, USA) and incubated in 30 mL of EM containing 0.1% DMSO alone, or 0.1% DMSO with either 250 µM cisplatin or 1 mM cisplatin. DMSO and cisplatin-treated groups were treated for 2 h. After completion of treatment, all groups were rinsed three times in EM and allowed to recover in 30 mL EM for 24 h. 0.1% DMSO was used as a vehicle and negative control. After completion of the treatment protocol, individual larvae (7 dpf) were placed at the front of a translucent microflume containing EM and then situated above a panel of LED lights emitting infrared (IR) light at 850 nm. Their swimming behavior was recorded in darkness with a high-speed IR camera (SC1 Edgertronic, Sanstreak Corp, Campbell, CA, USA) in the presence or absence of water flow generated by an Arduino-controlled water pump (6 V bow thruster motor, 108-01, Raboesch, Zeewolde, The Netherlands). A machine learning algorithm developed in DeepLabCut, a 3D markerless pose estimation program, was used to track the swim bladder of larvae within the microflume. Two-dimensional spatial heat maps of x- and y-axis larval positioning over time were generated in R (version 4.0.3). 

### 4.10. Live Hair Cell Labeling, Imaging, and Analysis

Live imaging was performed on wildtype zebrafish to investigate the effect of cisplatin ototoxicity on the function of surviving hair cells. DAPI (4′,6-diamidino-2-phenylindole, Sigma Aldrich, Darmstadt, Germany) was used as a nuclear stain to identify living hair cells and FM1-43 (N-(3-Triethylammoniumpropyl)-4-(4-(Dibutylamino) Styryl) Pyridinium Dibromide, ThermoFisher, Waltham, MA, USA) was used to evaluate functioning mechanotransduction in surviving hair cells. After undergoing treatment as described in the rheotaxis behavioral assay section, groups of ~5–10 larvae (7 dpf) were placed in 25 mm cell strainers (Corning Cell Strainer, Corning, NY, USA) and transferred to a 6-well plate with 8 mL EM containing 14 µM DAPI and 3 µM FM1-43. The larvae were co-incubated with these vital dyes for 20 s in the dark, rinsed two times in EM, and anesthetized in 30 mL EM containing 0.01% tricaine. 

Following sedation, larvae underwent live imaging as previously described with the following changes. An open bath imaging chamber (Warner Instruments; RC-26G; Holliston, MA, USA) was filled with 1 mL EM containing 0.01% tricaine and loaded with 3–4 larvae. The larvae were secured in place with a slice anchor (Warner Instruments; 64-0253, Holliston, MA, USA) lateral-side up. Single-channel z-stack images (1 µm step size; ~25 steps per Z-stack) of neuromasts L2–L4 were taken with a Hamamatsu ORCA-Flash 4.0 V3 camera using an X-Light V2TP spinning disk confocal and a 63x/0.9N.A. water immersion objective on a Leica DM6 Fixed Stage microscope. DAPI imaging was performed using a 405 nm wavelength laser at 20% power with a 100 ms/frame exposure time. RFP imaging was achieved using a 555 nm wavelength laser at 20% power with a 100 ms/frame exposure time. Images were acquired using MetaMorph software (Molecular Devices, San Jose, CA, USA). To limit variability in signal decay between treatment groups, exposure to vital dyes were staggered such that imaging for each condition was completed within 30 min after exposure.

ImageJ software was used to process the acquired images. Background subtraction was performed using a rolling ball radius of 100 pixels for each FM1-43 and DAPI z-stack and maximum projection images were generated. Intact hair cells, defined as those with non-pyknotic DAPI-stained nuclei, were counted and then outlined with a circular region of interest (4 µM diameter) to measure the mean fluorescent pixel intensity of FM1-43. Data from individual hair cells were normalized to the median intensity of hair cells from the same fish. Non-viable hair cells, defined as those without DAPI nuclear staining or those containing pyknotic nuclei, were excluded from analysis.

### 4.11. Statistics

Statistical analyses were performed with Prism 9 (GraphPad Software Inc., San Diego, CA, USA). Datasets were confirmed for normality using the D’Agostino-Pearson test. Statistical significance between two groups was determined using an unpaired Student’s *t*-test or a Mann–Whitney U test, as appropriate. Comparison of multiple groups was evaluated by one-way ANOVA or Kruskal–Wallis test with appropriate post hoc tests. For datasets dependent on multiple independent variables, statistical significance was determined using two-way ANOVA and appropriate post hoc tests.

## Figures and Tables

**Figure 1 ijms-23-14302-f001:**
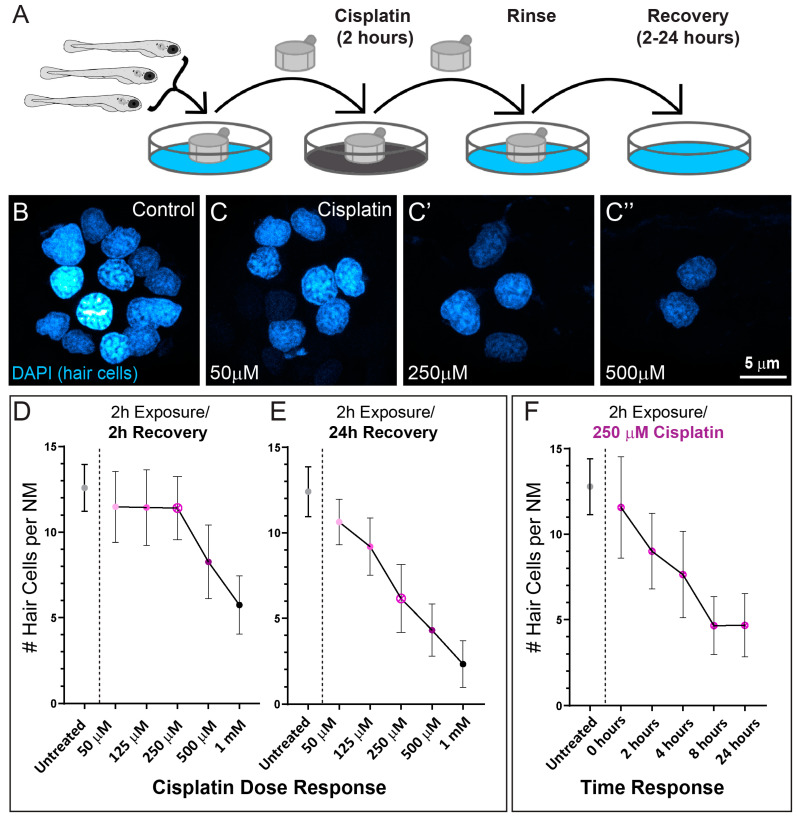
Dose- and time- response relationships of lateral line hair cell loss following 2 h exposure to cisplatin. (**A**) Diagram of standardized larval zebrafish cisplatin exposure protocol. Larvae (6 dpf) were placed into cell strainers and moved between cisplatin exposure and EM rinse steps, while being allowed to swim freely in EM during recovery. (**B**,**C**) Maximum intensity projection images of neuromast hair cell nuclei labeled with DAPI in either control (**B**) or cisplatin-treated (**C**–**C”**) larvae following 24 h recovery. (**D**,**E**) Cisplatin dose response relationship of hair cell loss per neuromast following either 2 h (**D**) or 24 h (**E**) recovery. Each dot represents the mean number of hair cells from 47–50 neuromasts/3 experimental trials. Note the delay in maximal hair cell loss at all doses examined. (**F**) Time response relationship following exposure to 250 µM cisplatin. Each dot represents the mean number of hair cells from 16–20 neuromasts. Maximal hair cell loss appeared to occur between 4–8 h following cisplatin exposure. Bars = SD.

**Figure 2 ijms-23-14302-f002:**
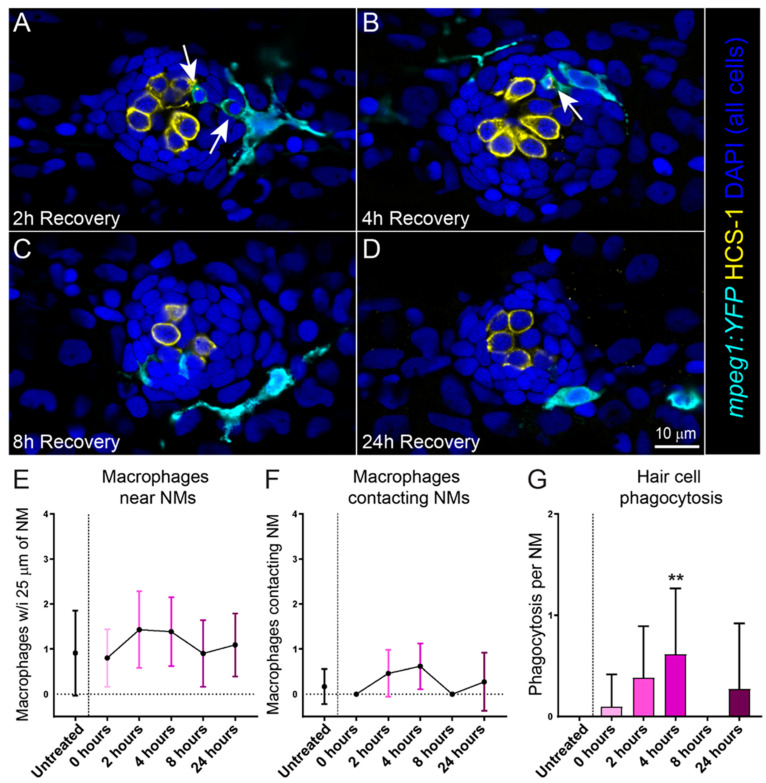
Macrophages respond to cisplatin-induced injury and phagocytose dying hair cells. (**A**–**D**) Single z-section images taken from 15 µm-depth confocal stacks of neuromasts in fish following treatment with 250 µM cisplatin. Macrophages are labeled with YFP (cyan), hair cells with an antibody to Otoferlin (HCS-1; yellow) and all cell nuclei are labeled with DAPI (blue). Arrows indicate macrophage phagocytosis of dying hair cells. Note that the macrophages at 2- and 4-h recovery (**A**,**B**) have internalized otoferlin-labeled hair cell debris (yellow) as well as pyknotic nuclei (blue). At later time points (**C**,**D**) macrophages are in proximity of neuromasts but no longer phagocytosing hair cell debris (**E**,**F**). Quantification of macrophages in proximity to neuromasts (**E**) and contacting neuromasts (**F**) following cisplatin. There was no observed difference between cisplatin treated and control. (**G**) Quantification of phagocytotic events in cisplatin treated neuromasts, which peaked 4 h post cisplatin treatment (Dunn’s multiple comparisons test, ** adjusted *p* = 0.0636). Bars = SD. Data obtained from 10–14 fish/treatment group.

**Figure 3 ijms-23-14302-f003:**
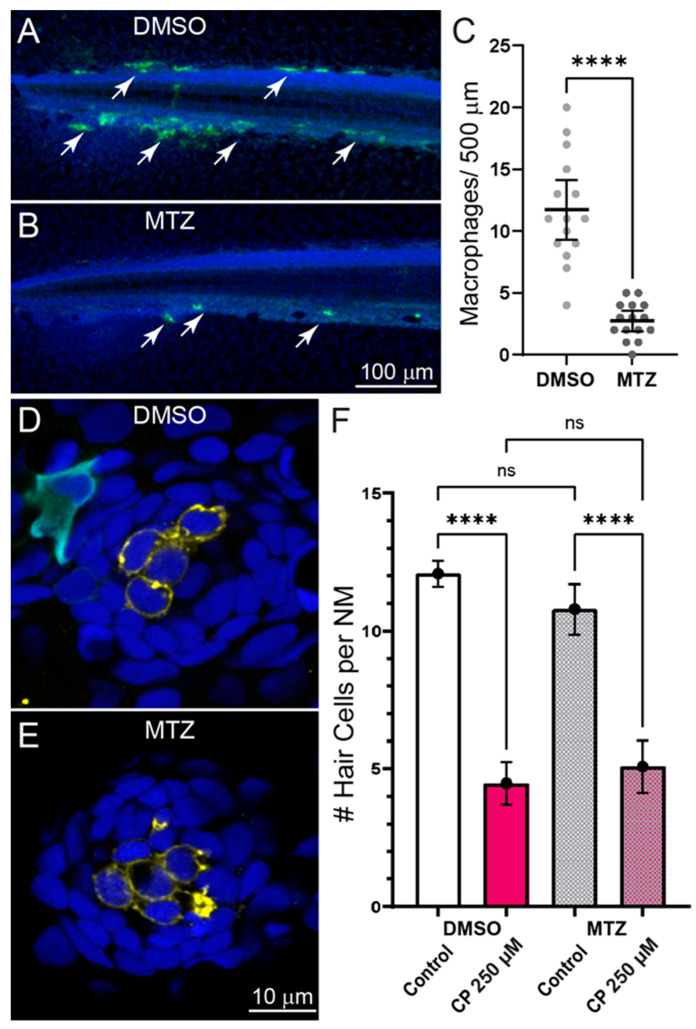
Depleting macrophages does not affect cisplatin induced neuromast hair cell loss. (**A**–**C**) Selective depletion of macrophages in double transgenic fish (gl25Tg, *Tg(mpeg1.1:GAL4FF*); c264Tg, *Tg(UAS-E1B:NTR-mCherry*). (**A**,**B**) Representative images of macrophage distribution (indicated with arrows) within the posterior-most 500 µm of the spinal column of fish treated with for 24 h with 0.1% DMSO (**A**) or 10 mM MTZ (**B**). (**C**) MTZ-treated fish showed a significant depletion of macrophages relative to DMSO-treated control (Unpaired *t*-test; **** *p* < 0.0001). Bars = Mean w/95% CI. Data obtained from 16 fish/treatment group. (**D**,**E**) Single z-section images taken from confocal stacks of neuromasts in fish following treatment with 250 µM cisplatin. Macrophages are labeled with YFP (cyan), hair cells with an antibody to Otoferlin (HCS-1; yellow) and all cell nuclei are labeled with DAPI (blue). (**F**) Quantification of hair cell loss following exposure to 250 µM cisplatin and 24 h recovery. Significant hair cell loss following cisplatin was observed in both DMSO and MTZ treatment groups relative to controls (Tukey’s multiple comparisons test, **** adjusted *p* < 0.0001), but no difference in cisplatin-induced hair cell loss was observed between DMSO and MTX treatment groups (adjusted *p* = 0.6415) Bars = Mean w/95% CI. n = 37–39 fish per treatment group, N = 3 trials.

**Figure 4 ijms-23-14302-f004:**
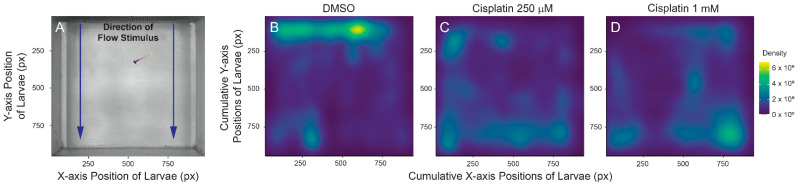
Cisplatin impairs larval ability to station hold in the presence of flow stimulus. (**A**) Top-down view of microflume with overlaying Cartesian coordinate system. Blue arrows indicate direction of water flow. Zero on the y axis indicates area of microflume with the strongest flow. Fish included for scale. (**B**–**D**) Two-dimensional spatial heat maps of cumulative positioning during flow stimulus indicate that lesioned fish accumulate towards the back of the microflume while DMSO-treated fish successfully station hold against current. n = 8–10 fish. N = 6 trials.

**Figure 5 ijms-23-14302-f005:**
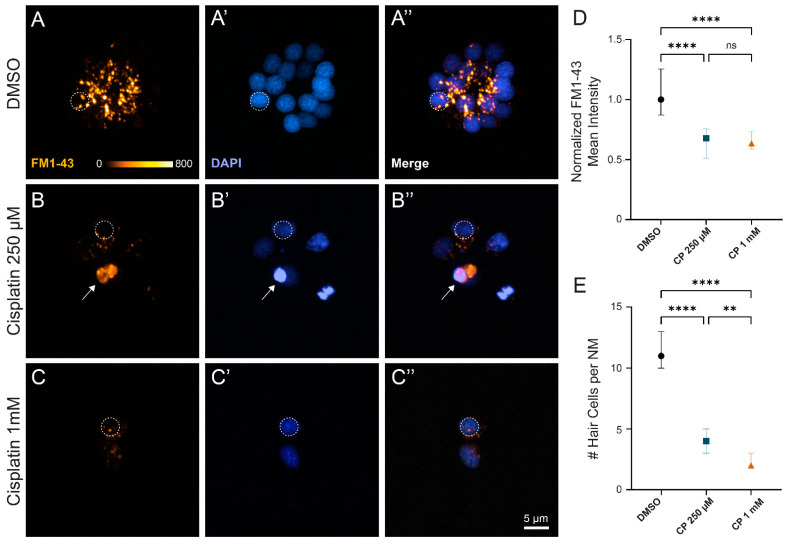
Surviving neuromast hair cells after cisplatin treatment demonstrate impaired mechanotransduction. (**A**–**C**) Maximum-intensity projections of confocal images show wildtype zebrafish neuromasts after 2 h treatment with 0.1% DMSO (**A**), 250 µM cisplatin (**B**), and 1 mM cisplatin (**C**) followed by a 24 h recovery in EM. Dotted circles exemplify 4 µm region of interest used to measure mean FM1-43 intensity. Arrow indicates non-viable hair cell that was not included in analysis (**B**). Mean hair cell FM1-43 intensity (normalized to control) demonstrates decreased mechanotransduction among cisplatin-treated fish compared to those exposed to vehicle control ((**D**), **** *p* < 0.0001, Kruskal–Wallis test). Post hoc analysis with Dunnett’s multiple comparisons test failed to detect significant changes in FM1-43 uptake between moderate and high dose cisplatin ((**D**), ns > 0.9999). Cisplatin demonstrates a dose-dependent increase in hair cell loss ((**E**), **** *p* < 0.0001, ** *p* = 0.0069; Dunnett’s multiple comparisons test). Median values with corresponding 95% confidence intervals shown for both FM1-43 uptake and hair cell counts. n = 3–4 fish (2–3 neuromasts per fish). N = 3 trials.

## Data Availability

The data that support the findings of this study are available from the corresponding author upon request. Instructions on the device and analysis used for the rheotaxis assay can be found on the Open Science Framework Repository: https://osf.io/rvyfz/ (accessed on 4 January 2022).

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
