# Peer review of "Evaluation of Cisplatin-Induced Pathology in the Larval Zebrafish Lateral Line"

_ijms, 2022, doi:10.3390/ijms232214302_

Round 1

Reviewer 1 Report

Lee et al. reported a method for the use of zebrafish larva in studies of cisplatin ototoxicity. There has been full of research in this field, but the authors can still find something new and necessary/better in the method establishment. For example, shorter exposure time (2 hours) was enough to cause hair cell death with fewer off-target effects. Evaluation of residual hair cell function following cisplatin exposure is necessary, et al. The authors stated that their research provide guidance on standardizing methods for the use of zebrafish model in studies of cisplatin ototoxicity. I think this word was very good and provided more evidence for the optimization of standardizing methods. However major revisions are needed.

1.      I just want to know how DAPI can specifically label the hair cells in the neuromast? Are there any other types of cells?

2.      Fig 1B and 1C-C’’ are figures from 2h treatment and 24h recovery, but they were not properly stated in the 2.1 section.

3.      Fig 3, MTZ cause macrophage ablation in the transgenic zebrafish line, but if MTZ (in the same dosage and the same exposure frame) can cause hair cell loss in the wild-type fish? The authors should exclude that the hair cell loss in transgenic line was caused by the additional effect of CP 250 μM and MTZ co-culture.

4.      Line 200, “Supplemental videos 1-4” should be “Supplemental videos 1-3”. There are only three videos in the supplemental materials.

5.      The authors stated that they want to provide guidance on standardizing methods for the use of zebrafish model in studies of cisplatin ototoxicity. There are already lots of similar studies on this field as mentioned in the discussion part, but although these studies have different cisplatin exposure time frame and dosage, or even different phenotype detection methods, but whether these discrepancies in the ototoxicity model establishment have impact on the otoprotective compounds screening? I think it’s better to mention or speculate in the discussion part.

Author Response

Lee et al. reported a method for the use of zebrafish larva in studies of cisplatin ototoxicity. There has been full of research in this field, but the authors can still find something new and necessary/better in the method establishment. For example, shorter exposure time (2 hours) was enough to cause hair cell death with fewer off-target effects. Evaluation of residual hair cell function following cisplatin exposure is necessary, et al. The authors stated that their research provide guidance on standardizing methods for the use of zebrafish model in studies of cisplatin ototoxicity. I think this word was very good and provided more evidence for the optimization of standardizing methods. However major revisions are needed.

  1. I just want to know how DAPI can specifically label the hair cells in the neuromast? Are there any other types of cells?

In live zebrafish larvae, DAPI preferentially enters hair cell through their mechanoelectrical transduction (MET) channels, which are large mechanically gated ion channels unique to hair cell that open upon deflection. We have added this information to methods section 4.2. In addition, as we have previously published use of this technique to specifically label lateral line hair cells, we have added this citation to methods section.

  1. Fig 1B and 1C-C’’ are figures from 2h treatment and 24h recovery, but they were not properly stated in the 2.1 section.

We have revised section 2.1 to correspond to the elements of Fig 1 more accurately. 

  1. Fig 3, MTZ cause macrophage ablation in the transgenic zebrafish line, but if MTZ (in the same dosage and the same exposure frame) can cause hair cell loss in the wild-type fish? The authors should exclude that the hair cell loss in transgenic line was caused by the additional effect of CP 250 μM and MTZ co-culture.

We do not see any hair cell loss when fish are exposed to MTZ alone Fig 3F; gray bar). In addition, we see no increase in hair cell loss when fish are pretreated with MTZ prior to cisplatin vs. cisplatin alone (Fig 3F).

  1. Line 200, “Supplemental videos 1-4” should be “Supplemental videos 1-3”. There are only three videos in the supplemental materials.

Thank you for catching that error—it has been corrected.

  1. The authors stated that they want to provide guidance on standardizing methods for the use of zebrafish model in studies of cisplatin ototoxicity. There are already lots of similar studies on this field as mentioned in the discussion part, but although these studies have different cisplatin exposure time frame and dosage, or even different phenotype detection methods, but whether these discrepancies in the ototoxicity model establishment have impact on the otoprotective compounds screening? I think it’s better to mention or speculate in the discussion part.

We detailed our guidance for standardized methods for the use of zebrafish model in studies of cisplatin ototoxicity by adding the following statements to our discussion section (added text is in bold):

Line 272-280: “Therefore, we propose that future investigations of cisplatin ototoxicity in zebrafish larvae use a standardized 2-hour exposure protocol laid out in Fig 1 A to minimize cisplatin’s non-specific toxicities while still capturing maximal hair cell loss. This protocol uses a broad range of cisplatin concentrations (50 µM to 1 mM) and recovery periods (2 to 24 hours) to accommodate different research questions, with corresponding dose- and time-response curves in Fig 1 D – F as guides. Such a standard exposure protocol may facilitate comparisons across different studies of cisplatin ototoxicity and otoprotective compounds.

Line 310-317: “Consequently, we encourage evaluation of residual hair cell function following cisplatin exposure in zebrafish to complement morphometric analysis in a similar way that distortion product otoacoustic emissions and auditory brain response experiments support morphometric analysis in the murine model. In the context of novel drug discovery, we speculate that performing morphometric hair cell analysis as a screening readout for otoprotective compounds and then corroborating positive results with a functional assay may increase successful translation to other preclinical models.”

Reviewer 2 Report

I found this manuscript to be interesting and well presented.  The data presented are convincing and the conclusions sound.  The manuscript could be improved by following up ideas I had and were in fact presented in the discussion.  What happens to these fish as they mature?  Are there any changes in the neuromasts beyond 24 hrs?  Do the fish survive?  Finally, can the methods shown be shown to actually measure rescue by interventions with drugs?

Author Response

I found this manuscript to be interesting and well presented.  The data presented are convincing and the conclusions sound.  The manuscript could be improved by following up ideas I had and were in fact presented in the discussion.

What happens to these fish as they mature?  

We did not explore this question in this study. It is unclear whether function, as defined by FM1-43 or rheotaxis behavior, recovers. Since zebrafish hair cells can regenerate 48-72 hours following injury (beyond the time course that we examined for this study), we speculate that lateral-line function would return to its baseline over time. This is an interesting question we would like to explore in future studies.

Are there any changes in the neuromasts beyond 24 hrs?

We stopped at 24 hrs to capture phenotypic changes prior to any spontaneous hair cell regeneration. We have noted this detail in Methods section 4.3.

Do the fish survive?

All the fish survive cisplatin exposure, even at higher doses. We speculate that the relatively brief exposure time to cisplatin limits off target effects that could be lethal. Current experiments in our lab are exploring the time course of regeneration following cisplatin exposure.

Finally, can the methods shown be shown to actually measure rescue by interventions with drugs?

We anticipate that these methods will provide more useful information on the ability of drugs to provide protection from cisplatin ototoxicity than the current standard of only quantifying hair cell loss. Assessing the protective effects of known and candidate protective drugs against cisplatin ototoxicity using these techniques is the goal of a follow-up study.  

Reviewer 3 Report

The manuscript entitled “Evaluation of cisplatin-induced pathology in the larval zebrafish lateral line” by David S Lee and coworkers describes a procedure to evaluate cisplatin ototoxicity. The authors show that cisplatin initiates progressive lateral hair cell loss, even after short exposure times, investigate the inflammatory response and describe a rheotaxis assay to evaluate residual hair loss function after cisplatin exposure of zebrafish larvae.

The manuscript is clear, very well written and presented. The experiments are well conducted. The materials and methods are greatly detailed. The conclusions are supported by the data presented.

The overall quality of the manuscript is good and I would suggest only few points as a way of improvement:

(1) Figure 5: It should be indicated that A’’, B’’ and C’’ correspond to merge of A and A’, B and B’, and C and C’, respectively.

(2) It is not clear whether within a given zebrafish larvae cisplatin exposure alters in the same way all the neuromasts of the lateral line. Could the authors state on that?

(3) At several places of the manuscript (Lines 17, 69, 243, …), the authors indicate that one goal of their study is to produce standardized protocols. However, they do not describe such a procedure. Could the authors be more precise about the standardizing way or alternatively understate this objective?

Author Response

The manuscript entitled “Evaluation of cisplatin-induced pathology in the larval zebrafish lateral line” by David S Lee and coworkers describes a procedure to evaluate cisplatin ototoxicity. The authors show that cisplatin initiates progressive lateral hair cell loss, even after short exposure times, investigate the inflammatory response and describe a rheotaxis assay to evaluate residual hair loss function after cisplatin exposure of zebrafish larvae.

The manuscript is clear, very well written and presented. The experiments are well conducted. The materials and methods are greatly detailed. The conclusions are supported by the data presented.

The overall quality of the manuscript is good and I would suggest only few points as a way of improvement:

(1) Figure 5: It should be indicated that A’’, B’’ and C’’ correspond to merge of A and A’, B and B’, and C and C’, respectively.

We have added the label “Merge” on the bottom left corner of Figure 5 A”.

(2) It is not clear whether within a given zebrafish larvae cisplatin exposure alters in the same way all the neuromasts of the lateral line. Could the authors state on that?

We observed comparable hair cell loss following cisplatin exposures in anterior (SO1, SO2 and SO3) and posterior (L3-5) neuromasts.

(3) At several places of the manuscript (Lines 17, 69, 243, …), the authors indicate that one goal of their study is to produce standardized protocols. However, they do not describe such a procedure. Could the authors be more precise about the standardizing way or alternatively understate this objective?

We detailed our guidance for standardized methods for the use of zebrafish model in studies of cisplatin ototoxicity. which we show in Fig. 1 A, by adding the following statements to our discussion section (added text is in bold):

Line 272-280: “Therefore, we propose that future investigations of cisplatin ototoxicity in zebrafish larvae use a standardized 2-hour exposure protocol laid out in Fig 1 A to minimize cisplatin’s non-specific toxicities while still capturing maximal hair cell loss. This protocol uses a broad range of cisplatin concentrations (50 µM to 1 mM) and recovery periods (2 to 24 hours) to accommodate different research questions, with corresponding dose- and time-response curves in Fig 1 D – F as guides. Such a standard exposure protocol may facilitate comparisons across different studies of cisplatin ototoxicity and otoprotective compounds.”

Line 310-317: “Consequently, we encourage evaluation of residual hair cell function following cisplatin exposure in zebrafish to complement morphometric analysis in a similar way that distortion product otoacoustic emissions and auditory brain response experiments support morphometric analysis in the murine model. In the context of novel drug discovery, we speculate that performing morphometric hair cell analysis as a screening readout for otoprotective compounds and then corroborating positive results with a functional assay may increase successful translation to other preclinical models.”

Additionally, we have added for clarity in Fig. 1 A “(2 – 24 hours)” under Recovery.